# Requirement for YAP1 signaling in myxoid liposarcoma

Marcel Trautmann[1,2,†] (ID), Ya-Yun Cheng[3,4,†], Patrizia Jensen[3,4], Ninel Azoitei[5], Ines Brunner[3], Jennifer Hüllein[4,6] (ID), Mikolaj Slabicki[6], Ilka Isfort[1,2] (ID), Magdalene Cyra[1,2] (ID), Ruth Berthold[1,2] (ID), Eva Wardelmann[1], Sebastian Huss[1], Bianca Altvater[7] (ID), Claudia Rossig[7,8] (ID), Susanne Hafner[9], Thomas Simmet[9], Anders Ståhlberg[10,11,12] (ID), Pierre Åman[10] (ID), Thorsten Zenz[6,13] (ID), Undine Lange[14], Thomas Kindler[14,15], Claudia Scholl[15,16], Wolfgang Hartmann[1,2,*,‡] (ID) & Stefan Fröhling[3,15,‡,**] (ID)

## Abstract

Myxoid liposarcomas (MLS), malignant tumors of adipocyte origin, are driven by the *FUS-DDIT3* fusion gene encoding an aberrant transcription factor. The mechanisms whereby FUS-DDIT3 mediates sarcomagenesis are incompletely understood, and strategies to selectively target MLS cells remain elusive. Here we show, using an unbiased functional genomic approach, that FUS-DDIT3-expressing mesenchymal stem cells and MLS cell lines are dependent on YAP1, a transcriptional co-activator and central effector of the Hippo pathway involved in tissue growth and tumorigenesis, and that increased YAP1 activity is a hallmark of human MLS. Mechanistically, FUS-DDIT3 promotes YAP1 expression, nuclear localization, and transcriptional activity and physically associates with YAP1 in the nucleus of MLS cells. Pharmacologic inhibition of YAP1 activity impairs the growth of MLS cells *in vitro* and *in vivo*. These findings identify overactive YAP1 signaling as unifying feature of MLS development that could represent a novel target for therapeutic intervention.

**Keywords** FUS-DDIT3; Hippo pathway; myxoid liposarcoma; verteporfin; YAP1
**Subject Category** Cancer

See also: **C Regina & S Hettmer** (May 2019)

## Introduction

Myxoid liposarcomas (MLS) account for 5–10% of soft-tissue sarcomas and approximately 20% of malignant adipocytic tumors (Fletcher *et al*, 2013). In the majority of cases, MLS arises in younger adults, thus representing the most frequent liposarcoma subtype in patients below the age of 20 years. Clinically, MLS are characterized by a high rate of local recurrence and development of distant metastases in approximately 40% of patients (Dei Tos, 2014). Morphologically, MLS comprise a broad spectrum of subtypes ranging from paucicellular myxoid tumors to hypercellular, round-cell high-grade sarcomas associated with a more aggressive clinical course (Antonescu *et al*, 2001). Genetically, the vast majority of MLS are characterized by a t(12;16)(q13;p11) chromosomal translocation that juxtaposes parts of the *FUS* gene to the entire coding sequence of *DDIT3*. The resulting FUS-DDIT3 fusion protein, which acts as a transcriptional (dys-)regulator, has been shown to play an essential role in MLS pathogenesis (Kuroda *et al*, 1997;

1 Gerhard-Domagk-Institute of Pathology, Münster University Hospital, Münster, Germany
2 Division of Translational Pathology, Gerhard-Domagk-Institute of Pathology, Münster University Hospital, Münster, Germany
3 Department of Translational Medical Oncology, National Center for Tumor Diseases (NCT) Heidelberg and German Cancer Research Center (DKFZ), Heidelberg, Germany
4 Faculty of Biosciences, Heidelberg University, Heidelberg, Germany
5 Department of Internal Medicine I, Ulm University Hospital, Ulm, Germany
6 Department of Translational Oncology, National Center for Tumor Diseases (NCT) Heidelberg and German Cancer Research Center (DKFZ), Heidelberg, Germany
7 Department of Pediatric Hematology and Oncology, University Children's Hospital Münster, Münster, Germany
8 Cells in Motion Cluster of Excellence, University of Münster, Münster, Germany
9 Institute of Pharmacology of Natural Products and Clinical Pharmacology, Ulm University Hospital, Ulm, Germany
10 Department of Pathology and Genetics, Sahlgrenska Cancer Center, Institute of Biomedicine, Sahlgrenska Academy at University of Gothenburg, Gothenburg, Sweden
11 Wallenberg Centre for Molecular and Translational Medicine, University of Gothenburg, Gothenburg, Sweden
12 Department of Clinical Pathology and Genetics, Sahlgrenska University Hospital, Gothenburg, Sweden
13 Department of Hematology, Zurich University Hospital and University of Zurich, Zürich, Switzerland
14 Department of Hematology, Medical Oncology and Pneumology, University Medical Center of Mainz, Mainz, Germany
15 German Cancer Consortium, Heidelberg (Frankfurt/Mainz), Germany
16 Division of Applied Functional Genomics, DKFZ, Heidelberg, Germany
  *Corresponding author. Tel: +49 251 83 58479; Fax: +49 251 83 55481; E-mail: wolfgang.hartmann@ukmuenster.de
  **Corresponding author. Tel: +49 6221 56 35212; Fax: +49 6221 56 5389; E-mail: stefan.froehling@nct-heidelberg.de
  †These authors contributed equally to this work as first authors
  ‡These authors contributed equally to this work as senior authors

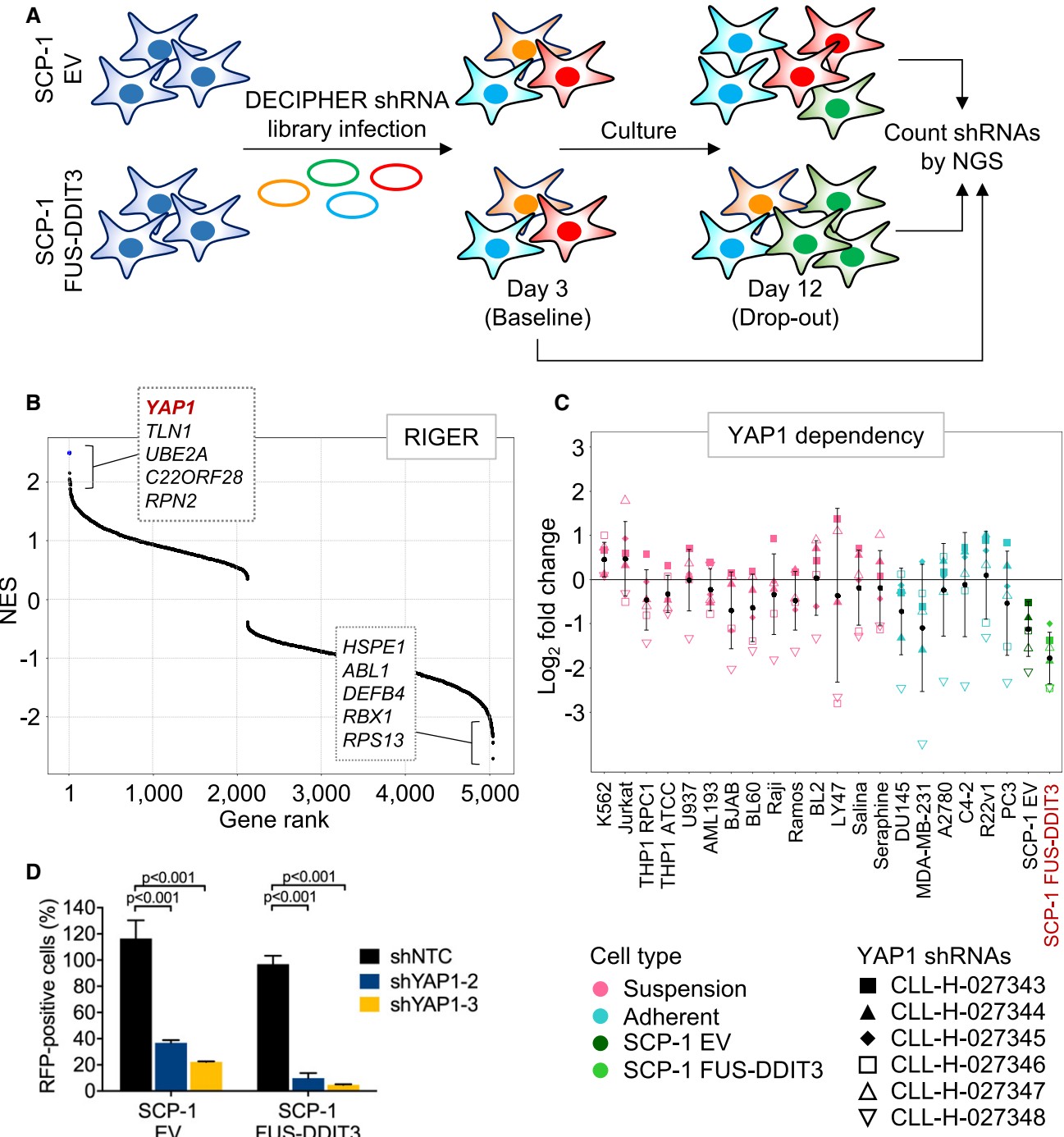

**Figure 1. Identification of genes required by *FUS-DDIT3*-expressing mesenchymal stem cells.**

A   Schematic of RNAi screens. SCP-1 cells expressing *FUS-DDIT3* or EV were transduced with Module 1 of the DECIPHER Pooled Lentiviral Human Genome-Wide shRNA Library. Half of the cells were harvested on day 3 (baseline sample) and day 12 (drop-out sample), respectively, and shRNA abundance was determined by next-generation sequencing (NGS).

B   RIGER analysis to identify genes that are preferentially essential in FUS-DDIT3-expressing SCP-1 cells. EV-transduced SCP-1 cells and 20 FUS-DDIT3-negative cancer cell lines screened with the same shRNA library were used as reference set. Genes were ranked according to relative shRNA depletion, and *YAP1* was identified as top FUS-DDIT3-specific essential gene. NES, normalized enrichment score.

C   LFC change in *YAP1* shRNA representation in 20 cancer cell lines and SCP-1 cells transduced with *FUS-DDIT3* or EV. Black dots and error bars represent the mean ± SD of LFC scores for six independent shRNAs.

D   Competition assays with SCP-1 cells transduced with RFP-labeled NTC or *YAP1* shRNAs. Flow cytometric quantification of RFP-positive cells on day 9 relative to day 3 showed that *YAP1* knockdown was preferentially toxic to *FUS-DDIT3*-expressing cultures. Bars and error bars represent the mean ± SD of two independent experiments, two-way ANOVA.

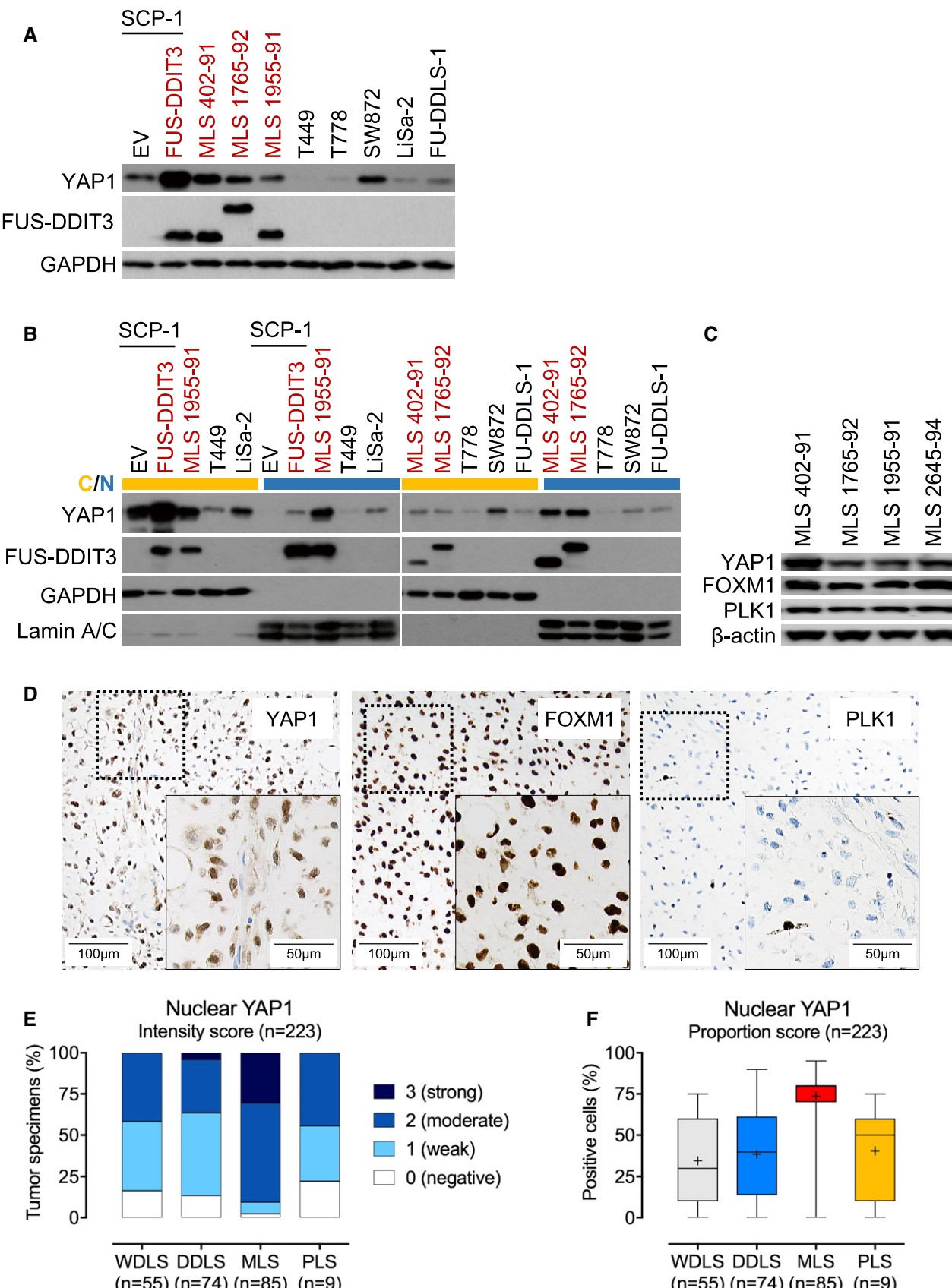

**Figure 2.**

◀ **Figure 2.  Increased nuclear YAP1 levels in FUS-DDIT3-expressing mesenchymal stem cells, MLS cell lines, and MLS patient samples.**

A    Expression of YAP1 in SCP-1 cells transduced with *FUS-DDIT3* or EV and liposarcoma cell lines. One of at least two independent experiments with similar results is shown. FUS-DDIT3-expressing cell types are indicated in red.

B    Expression of YAP1 in cytoplasmic (yellow) and nuclear (blue) fractions from SCP-1 cells transduced with *FUS-DDIT3* or EV and liposarcoma cell lines. One of at least two independent experiments with similar results is shown. FUS-DDIT3-expressing cell types are indicated in red.

C    Expression of FOXM1 and PLK1 in MLS cell lines. One of at least two independent experiments with similar results is shown.

D    Strong nuclear expression of YAP1, FOXM1, and PLK1 in MLS patient samples (original magnification, ×10 [inset, ×20]).

E    Intensity of nuclear YAP1 expression in liposarcoma patient samples. Immunoreactivity was assessed using a semi-quantitative score (0, negative; 1, weak; 2, moderate; and 3, strong) defining the staining intensity in the positive control (hepatocellular carcinoma) as strong. Only tumors with at least moderate staining (semi-quantitative score ≥ 2) and ≥ 30% YAP1-positive cells were considered positive for the purposes of the study.

F    Proportion of cells with nuclear YAP1 expression in liposarcoma patient samples. Boxes represent mean values and lower and upper quartiles. Whiskers represent minimum and maximum values.

Perez-Losada *et al*, 2000; Engstrom *et al*, 2006; Riggi *et al*, 2006), but its mode of action remains incompletely understood.

Long-term survival in MLS patients may be achieved through radical surgery and adjuvant radiation and/or conventional chemotherapy (Jones *et al*, 2005; Ratan & Patel, 2016, 2017). Although MLS are more sensitive to cytotoxic agents than other liposarcoma subtypes, patients with disseminated disease are usually incurable and chemotherapy is generally administered with palliative intent, underlining the need for novel, biology-guided therapeutic options. MLS belong to the group of translocation-related sarcomas, which are characterized by "quiet" genomes with few mutations beyond the driving gene fusion. Therefore, counter-acting the effects of the chimeric FUS-DDIT3 oncoprotein represents, in principle, the most promising strategy to selectively target MLS cells. However, transcription factors are notoriously difficult to inhibit with small molecules, and FUS-DDIT3 has not yet been shown to be pharmacologically tractable.

In this study, we employed an unbiased functional genomic approach to search for signaling pathways that are selectively essential in cells expressing FUS-DDIT3. Large-scale RNA interference (RNAi) screening identified dependence on YAP1, a transcriptional co-activator that is physiologically inhibited by the Hippo pathway responsible for limiting tissue growth and organ size (Pan, 2010), as specific liability of FUS-DDIT3-expressing MLS cells that could be exploited for therapeutic benefit.

# Results

## RNAi screen for essential genes in FUS-DDIT3-expressing mesenchymal stem cells

To identify genes and/or cellular processes that are essential specifically in the context of the *FUS-DDIT3* fusion gene, we performed drop-out RNAi screens in two SCP-1 immortalized human mesenchymal stem cell lines (Bocker *et al*, 2008; Haasters *et al*, 2009) expressing *FUS-DDIT3* or empty vector (EV; Fig 1A). Screens were conducted using Module 1 of the DECIPHER Pooled Lentiviral Human Genome-Wide shRNA Library, which consists of approximately 27,500 shRNAs targeting over 5,000 human genes (Fig 1A, Appendix Fig S1). Candidates for further functional and mechanistic investigation were selected based on a stepwise approach. We first integrated the data obtained in SCP-1 cells with the results of previous DECIPHER screens conducted in cell lines representing a range of hematopoietic (Burkitt lymphoma, *n* = 8;

T-cell acute lymphoblastic leukemia, *n* = 1; acute myeloid leukemia, *n* = 4; chronic myeloid leukemia, *n* = 1) and epithelial (prostate cancer, *n* = 4; breast cancer, *n* = 1; ovarian cancer, *n* = 1) malignancies (Appendix Fig S1). Following normalization of shRNA depletion values using PMAD (Cheung *et al*, 2011; preprint: Huellein *et al*, 2018), we employed RIGER (Luo *et al*, 2008) to rank genes with respect to FUS-DDIT3-selective essential-ity and eliminated human core fitness genes identified in genome-wide CRISPR/Cas9 knockout screens (Shalem *et al*, 2014; Hart *et al*, 2015, 2017; Wang *et al*, 2015). This approach identified YAP1, a transcriptional co-activator and central effector of the Hippo pathway involved in embryonic development, tissue home-ostasis, and tumorigenesis (Zhao *et al*, 2011; Harvey *et al*, 2013), as lead hit with the highest normalized enrichment score (Fig 1B and C). The complete dataset obtained from the RNAi screen conducted in SCP-1 cells is provided as Dataset EV1. To validate the results of the shRNA screens, SCP-1 cells were lentivirally transduced with two different RFP-labeled shRNAs targeting *YAP1* or NTC shRNA at low transduction efficiency, resulting in mixed populations of transduced and untransduced cells. Flow cytometric analysis demonstrated that *YAP1* knockdown depleted RFP-positive cells preferentially in FUS-DDIT3-expressing cultures (Fig 1D, Appendix Fig S1).

## Increased YAP1 activity in MLS cell lines

To translate our findings in genetically engineered SCP-1 cells to the setting of endogenous FUS-DDIT3 expression, we first deter-mined YAP1 mRNA and protein levels in a panel of human liposarcoma cell lines using quantitative RT–PCR and immunoblot-ting. YAP1 was expressed more strongly in FUS-DDIT3-positive MLS 402-91, MLS 1765-92, and MLS 1955-91 cells than in cell lines representing other liposarcoma subtypes (T449 and T778, WDLS; FU-DDLS-1, DDLS; LiSa-2, PLS) with the exception of SW872 cells (PLS), which showed similar YAP1 mRNA and protein levels as MLS 1765-92 and MLS 1955-91 (Fig 2A, Appendix Fig S2A). Frac-tionation experiments followed by immunoblotting as well as IF analysis demonstrated that YAP1 was primarily localized in the nucleus of MLS cells, indicating transcriptional activity, whereas in SW872 cells the majority of YAP1 protein was retained in the cyto-plasm (Fig 2B, Appendix Fig S2B). Furthermore, we detected strong expression of the YAP1 downstream targets FOXM1 and PLK1 in MLS cell lines (Fig 2C). Together, these observations demonstrated that cultured human MLS cells exhibit increased YAP1 activity.

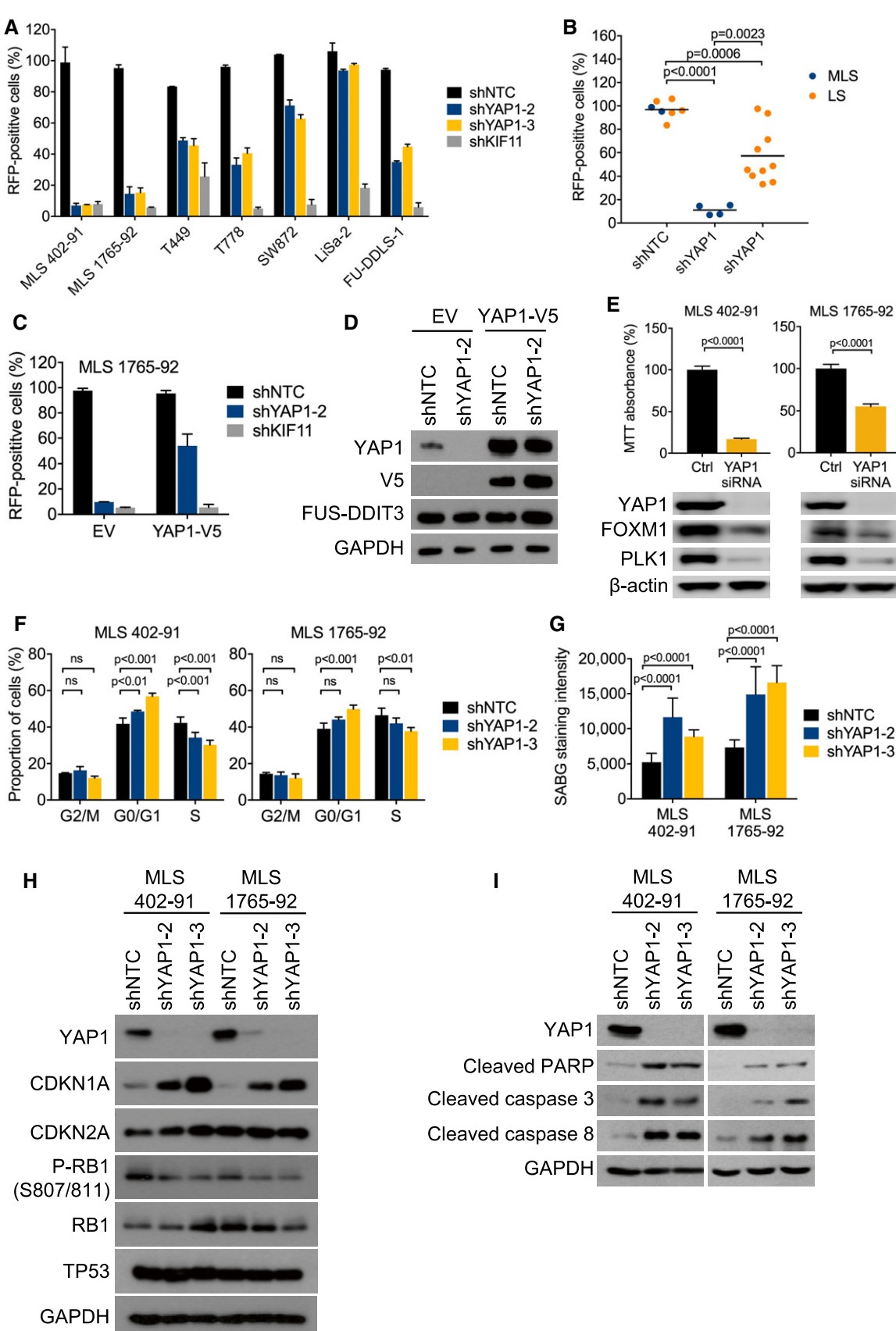

Figure 3.

**Figure 3. Requirement for YAP1 activity in MLS cell lines.**

A   Competition assays with liposarcoma cell lines transduced with RFP-labeled NTC or *YAP1* shRNAs. Flow cytometric quantification of RFP-positive cells on day 17 relative to day 3 showed that *YAP1* knockdown was preferentially toxic to MLS cells. Bars and error bars represent the mean $\pm$ SD of two independent experiments.

B   Aggregate data from competition assays shown in (A). Statistical significance was assessed using an unpaired *t*-test. LS, non-myxoid liposarcoma.

C   Competition assays with MLS 1765-92 cells transduced with an RFP-labeled NTC shRNA or an RFP-labeled shRNA against the *YAP1* 3′UTR following transduction with EV or the *YAP1* coding sequence. Flow cytometric quantification of RFP-positive cells on day 17 relative to day 3 showed that cell viability was rescued by expression of the shRNA-resistant *YAP1* cDNA. Bars and error bars represent the mean $\pm$ SD of two independent experiments.

D   Expression of total and exogenous V5-tagged YAP1 in MLS 1765-92 cells shown in (C). One of at least two independent experiments with similar results is shown.

E   Cell viability and expression of FOXM1 and PLK1 in MLS cell lines following siRNA-mediated *YAP1* knockdown. Bars and error bars represent the mean $\pm$ SD of three independent experiments, unpaired *t*-test. The blots represent one of at least three independent experiments with similar results.

F   Flow cytometric cell cycle analysis of MLS cell lines following shRNA-mediated *YAP1* knockdown. Bars and error bars represent the mean $\pm$ SD of three independent experiments, two-way ANOVA; ns, not significant.

G   Senescence-associated β-galactosidase (SABG) staining intensity in MLS cell lines following shRNA-mediated *YAP1* knockdown. Bars and error bars represent the mean $\pm$ SD of ten random microscopic fields, two-way ANOVA.

H   Expression of CDKN1A, CDKN2A, total and phosphorylated RB1, and TP53 in MLS cell lines following shRNA-mediated *YAP1* knockdown. One of at least two independent experiments with similar results is shown.

I   Expression of cleaved PARP and cleaved caspase-3/8 in MLS cell lines following shRNA-mediated *YAP1* knockdown. One of at least two independent experiments with similar results is shown.

### Increased YAP1 activity in MLS patient samples

To further explore the involvement of YAP1 in MLS development, we examined the expression of nuclear YAP1, corresponding to the transcriptionally active pool, in 223 primary human liposarcoma specimens (MLS, $n = 85$; WDLS, $n = 55$; DDLS, $n = 74$; PLS, $n = 9$) using IHC (Fig 2D–F). Among the MLS specimens, moderate to strong nuclear YAP1 levels were detected in 90.6% (77/85) of cases, whereas only 2.4% (2/85) of tumors displayed no YAP1 immunoreactivity; thus, according to the defined criteria, 90.6% of MLS tumors were positive for YAP1. Accordingly, we detected nuclear expression of FOXM1 and PLK1. In patients with WDLS, moderate nuclear YAP1 expression was detected in 41.8% (23/55) of cases, whereas 16.4% (9/55) of tumors showed no YAP1 staining. For DDLS, moderate to strong nuclear YAP1 staining was detected in 36.5% (27/74) of cases, while 13.5% (10/74) of specimens showed no nuclear YAP1 immunoreactivity. Among the PLS specimens, 44.4% (4/9) showed moderate nuclear YAP1 immunoreactivity, while 22.2% (2/9) of tumors were negative for nuclear YAP1. Nuclear expression of YAP1 in MLS specimens did not correlate with clinical characteristics such as patient age, gender, *FUS-DDIT3* transcript variant, or tumor size. These findings provided additional support that increased YAP1 activity represents a unifying feature in MLS.

### Requirement for YAP1 activity in MLS cell lines

To confirm the differential requirement for YAP1 identified by RNAi screen, we suppressed *YAP1* expression in seven human liposarcoma cell lines using two different shRNAs. *YAP1* knockdown depleted FUS-DDIT3-expressing MLS 402-91 and MLS 1765-92 cells to a similar extent as knockdown of *KIF11*, an essential cell cycle regulator that served as positive control, whereas there was little effect in cell lines representing other liposarcoma subtypes (Fig 3A and B, Appendix Fig S3A and B). To further ensure the specificity of these results, we performed rescue experiments with a shRNA targeting the 3′ untranslated region (UTR) of *YAP1* mRNA. We first transduced FUS-DDIT3-positive MLS 1765-92 cells with EV or the *YAP1* coding sequence, which lacks the 3′ UTR. Subsequent knockdown of endogenous *YAP1* inhibited the growth of EV-transduced cells, whereas the RNAi-induced phenotype was countered by expression of the shRNA-resistant *YAP1* cDNA (Fig 3C and D). In a

complementary approach, we observed that siRNA-mediated transient knockdown of *YAP1* also reduced the viability and proliferation of MLS cells, which was accompanied by decreased YAP1 target gene expression (Fig 3E). Together, these data indicated that FUS-DDIT3-positive human MLS cells are dependent on YAP1 activity.

### Proliferation arrest, senescence, and apoptosis by YAP1 suppression in MLS cells

To examine the functional basis for the depletion of FUS-DDIT3-expressing cells upon YAP1 suppression, we first analyzed the cell cycle profiles of MLS 402-91 and MLS 1765-92 cells 5 days after shRNA knockdown of *YAP1*. Flow cytometric analysis of EdU incorporation demonstrated that *YAP1* knockdown cells accumulated in the G1 phase of the cell cycle, whereas cells transduced with NTC shRNA were unaffected (Fig 3F, Appendix Fig S3C). In addition, *YAP1* knockdown resulted in a senescence-like phenotype, as evidenced by increased senescence-associated β-galactosidase activity (Fig 3G), loss of RB1 phosphorylation, and induction of CDKN1A (also known as p21$^{\text{cip1}}$) expression (Fig 3H). Finally, we observed that YAP1 suppression significantly increased apoptosis, as assessed by detection of caspase-3/8 and poly (ADPribose) polymerase (PARP) cleavage (Fig 3I).

### Causal relationship between FUS-DDIT3 expression and increased YAP1 activity

Our observations suggested a causal relationship between the presence of FUS-DDIT3 and increased YAP1 activity. Consistent with this hypothesis, stable expression of FUS-DDIT3 in SCP-1 cells increased *YAP1* mRNA levels approximately threefold compared to EV-transduced control cells (Appendix Fig S2A), which was paralleled by elevated total and nuclear YAP1 protein expression (Fig 2A and B). FUS-DDIT3 induced the expression and nuclear localization of YAP1 downstream effectors such as FOXM1 and PLK1 (Fig 4A and B). Furthermore, TEAD luciferase reporter assays employing the 8xGTIIC system demonstrated that expression of FUS-DDIT3 significantly increased YAP1-responsive luciferase activity compared to the backbone vector control (Fig 4C). Together, these findings supported a role for FUS-DDIT3 in establishing increased YAP1 activity in human MLS cells.

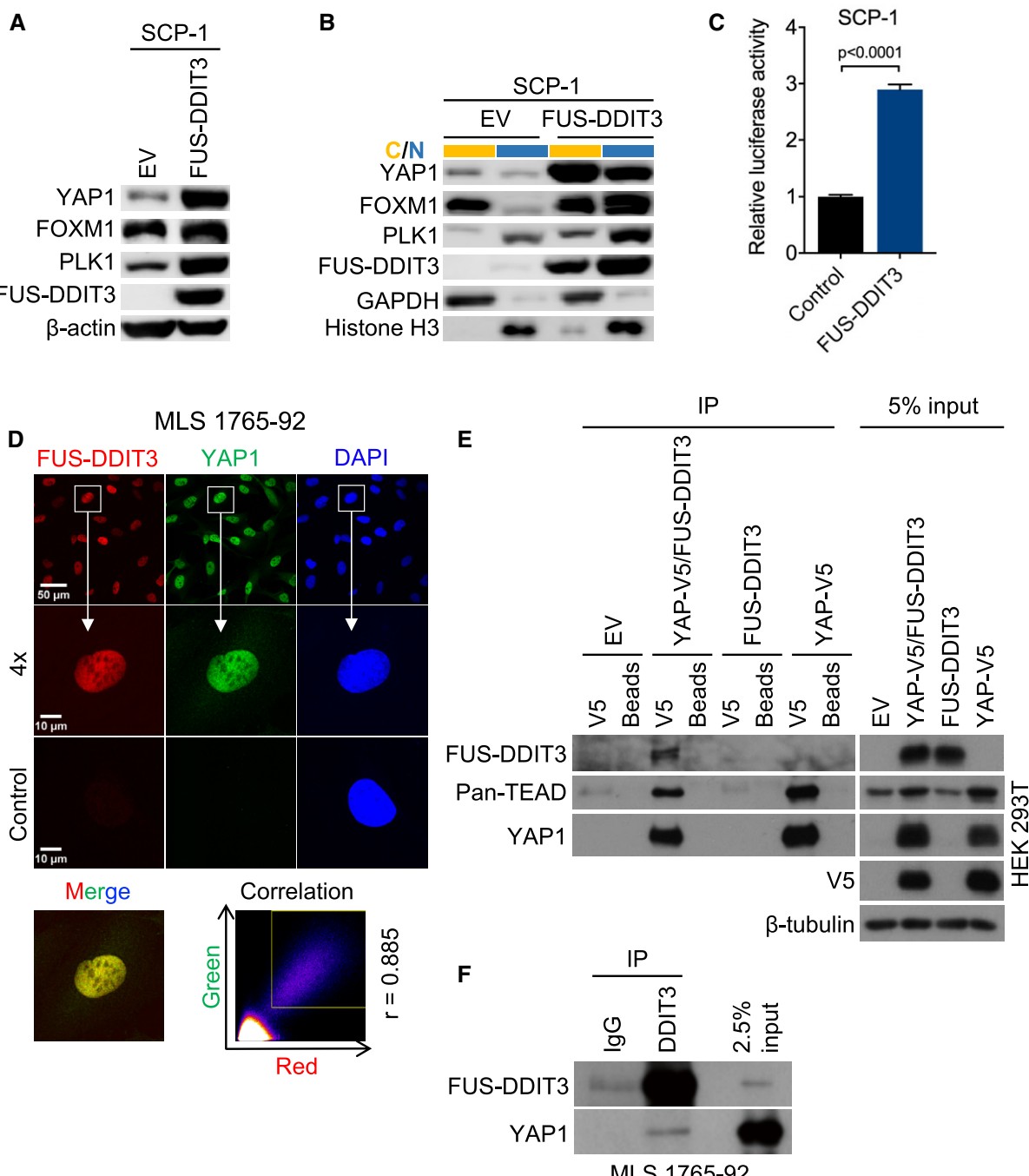

**Figure 4.   Causal relationship between FUS-DDIT3 expression and increased YAP1 activity.**

A   Expression of YAP1 and downstream effectors in SCP-1 cells transduced with *FUS-DDIT3* or EV. One of at least three independent experiments with similar results is shown.

B   Expression of YAP1 and downstream effectors in cytoplasmic (yellow) and nuclear (blue) fractions from SCP-1 cells transduced with *FUS-DDIT3* or EV. One of at least two independent experiments with similar results is shown.

C   YAP1-responsive luciferase activity in SCP-1 cells transduced with *FUS-DDIT3*. Relative luciferase activity is displayed as fold change relative to control. Bars and error bars represent the mean ± SD of three independent experiments, unpaired *t*-test.

D   Localization of FUS-DDIT3 and YAP1 in MLS 1765-92. Nuclei were counterstained with DAPI. The original magnification was ×63, and images were zoomed in four times for co-localization analysis. The correlation between red and green fluorescence was determined by Pearson coefficient analysis (square, area for signal acquisition).

E   Co-IP of transiently expressed FUS-DDIT3 and YAP1 from HEK293T cells. V5-tagged YAP1 was pulled down using an anti-V5 antibody, and interacting proteins were detected by immunoblotting. One of at least two independent experiments with similar results is shown.

F   Co-IP of endogenous FUS-DDIT3 and YAP1 from MLS 1765-92 cells. FUS-DDIT3 was pulled down using an anti-DDIT3 antibody, and interacting proteins were detected by immunoblotting. One of at least two independent experiments with similar results is shown.

Given the role of FUS-DDIT3 in deregulation of YAP1 expression and the predominant nuclear localization of YAP1 in MLS cells, we hypothesized that these proteins may interact to coordinately establish a gene expression program that promotes MLS development. In support of this concept, immunofluorescence demonstrated that FUS-DDIT3 and YAP1 co-localize in the nucleus of MLS 1765-92 and MLS 402-91 cells (Fig 4D, Appendix Fig S4), and their direct physical association was verified by co-IP of transiently expressed or endogenous proteins from HEK293T and MLS 1765-92 cells, respectively (Fig 4E and F).

### Sensitivity of MLS cells to pharmacologic inhibition of YAP1 activity

Our genetic data suggested that aberrant YAP1 activity might be a therapeutic target in MLS. We therefore evaluated the growth of MLS cells in the presence of verteporfin, a second-generation photosensitizer approved for the treatment of age-related macular degeneration that inhibits YAP1 signaling by disrupting the YAP1-TEAD complex and augmenting its sequestration in the cytoplasm (Liu-Chittenden et al, 2012; Brodowska et al, 2014; Wang et al, 2016). These experiments showed that verteporfin suppressed the viability and proliferation of all three MLS cell lines analyzed in a dose-dependent manner (Fig 5A, Table 1). The growth-inhibitory effects of verteporfin could be attributed to increased apoptosis and a significant reduction in mitotic activity, as assessed by flow cytometric quantification of cleaved PARP and phosphorylated histone H3$^{S10}$, respectively (Fig 5B, Table 2), and were accompanied by reciprocal dose-dependent changes in the expression of the YAP1 downstream effectors FOXM1 and PLK1 (Fig 5C) (Eisinger-Mathason et al, 2015; Fullenkamp et al, 2016). Finally, TEAD reporter assays demonstrated that verteporfin significantly decreased luciferase activity in MLS cell lines co-transfected with a constitutively active YAP1$^{S127A}$ mutant (Fig 5D). Collectively, these data showed that MLS cells are sensitive to pharmacologic blockade of YAP1 activity, indicating that their overreliance on YAP1 may provide a therapeutic opportunity.

### In vivo efficacy of YAP1 inhibition against MLS xenografts

To verify the effect of YAP1 inhibition on MLS growth in vivo, we deposited MLS 402-91 and MLS 1765-92 cells transduced with two different shRNAs targeting YAP1 on the surface of chicken CAM and observed that YAP1 knockdown significantly impaired their tumor-forming capacity (Fig 5E). In addition, topical administration of verteporfin to established MLS 402-91 and MLS 1765-92 xenografts resulted in a significant reduction of tumor volume compared to the vehicle-treated control group (Fig 5F). Collectively, these data showed that YAP1 inhibition impairs the initiation and maintenance of MLS tumors in vivo, further supporting the idea that overactive YAP1 signaling could represent a new target for therapeutic intervention in patients with FUS-DDIT3-driven MLS.

## Discussion

MLS is a lipogenic malignancy with propensity to local relapse and distant metastasis. High histological grade, defined as round cell component > 5%, serves as a major predictor of unfavorable

clinical outcome (Smith et al, 1996; Antonescu et al, 2001). Although high-grade MLS are more sensitive to conventional radio- and chemotherapy than other liposarcoma subtypes, prognosis in the metastatic situation is poor (Ratan & Patel, 2016). The chimeric FUS-DDIT3 fusion protein, a hallmark of MLS, acts as an aberrant transcription factor and has been shown to drive MLS development in mice (Kuroda et al, 1997; Riggi et al, 2006). Though FUS-DDIT3 has been documented to be incorporated into transcription complexes and to be associated with chromatin remodeling (Goransson et al, 2009), its specific mode of action remains incompletely understood. As other translocation-related sarcomas, e.g., Ewing sarcoma, synovial sarcoma, or alveolar rhabdomyosarcoma, MLS genomes harbor few somatic mutations, underscoring the dominant role of FUS-DDIT3 in MLS pathogenesis (Barretina et al, 2010; Crompton et al, 2014; Shern et al, 2014; Tirode et al, 2014; Trautmann et al, 2019). Since therapeutic inhibition of the chimeric fusion protein itself represents a challenge, it appears most promising to intercept signaling pathways that are functionally dependent on FUS-DDIT3 activity for selective targeting of MLS cells.

In this study, we employed an unbiased functional genomic approach to uncover that human mesenchymal stem cells engineered to express FUS-DDIT3 require YAP1, a transcriptional co-activator and central effector of the Hippo signal transduction pathway (Pan, 2010). The essential role of YAP1 in different epithelial malignancies is well established (Harvey et al, 2013); however, evidence implicating YAP1 in mesenchymal tumorigenesis is sparse (Crose et al, 2014; Tremblay et al, 2014; Eisinger-Mathason et al, 2015; Fullenkamp et al, 2016; Cancer Genome Atlas Research Network, 2017). To verify the requirement for YAP1 in the context of endogenous FUS-DDIT3 expression, we analyzed the prevalence and functional relevance of YAP1 in a representative panel of human liposarcoma cell lines and a large cohort of MLS tumor specimens. Expectedly, YAP1 positivity was not restricted to MLS; however, nuclear expression of YAP1 was significantly more prevalent in MLS compared to other liposarcomas, and YAP1-positive tumors showed strong expression of the YAP1 downstream targets FOXM1 and PLK1. These findings are in line with a previous immunohistochemical study (Fullenkamp et al, 2016) and provide evidence that increased YAP1-mediated transcriptional activity represents an essential feature of MLS. Accordingly, RNAi-based YAP1 depletion in MLS cells resulted in suppression of cell viability, cell cycle arrest, cellular senescence, and induction of apoptosis, accompanied by decreased YAP1 target gene expression.

Our observations in mesenchymal stem cells and MLS cell lines imply a functional link between FUS-DDIT3 expression and aberrant YAP1 activity, thereby providing new insights into the oncogenic properties of FUS-DDIT3. In addition to the induction of YAP1 transcription by FUS-DDIT3, our data also indicate a direct physical interaction between the FUS-DDIT3 and YAP1 proteins, which might point to the coordinate establishment of gene expression programs that promote MLS tumorigenesis. Given the contextual requirement for YAP1 activity, a YAP1-directed therapeutic approach could represent a rational strategy to selectively target FUS-DDIT3-expressing MLS cells. Consistent with this hypothesis, RNAi-mediated depletion of YAP1 or pharmacologic inhibition of the YAP1-TEAD transcriptional complex with verteporfin (Liu-Chittenden et al,

2012) suppressed cell viability and YAP1 target gene expression in MLS cells. The growth-suppressive effects of *YAP1* knockdown or verteporfin treatment could be recapitulated in MLS cell line-based xenograft models *in vivo*. Given recent reports that imbalances in proteostasis and subsequent proteotoxicity may also contribute to the anti-neoplastic activity of verteporfin (Zhang *et al*, 2015), the

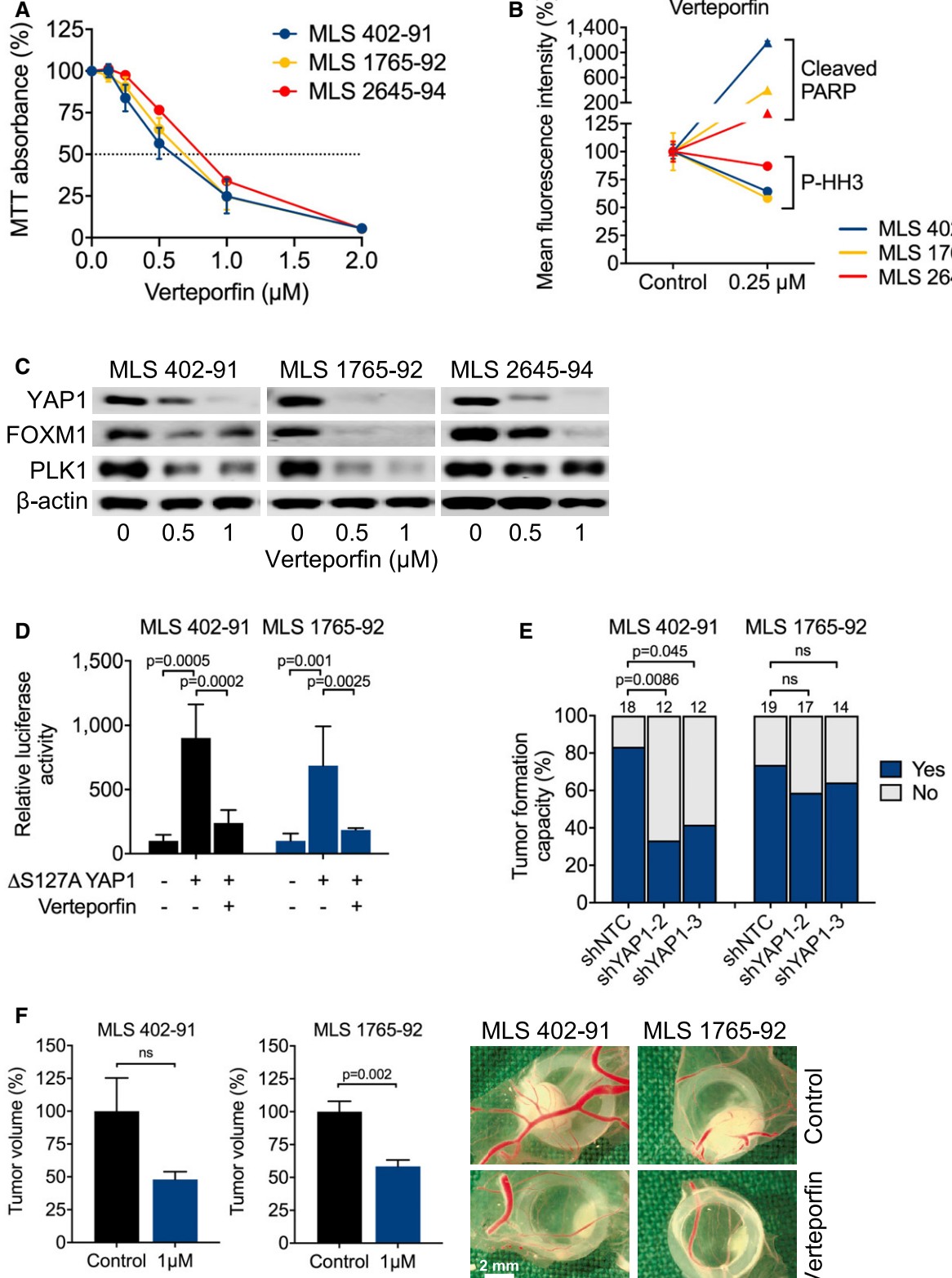

**Figure 5.**

**Figure 5.  Sensitivity of MLS cell lines to pharmacologic YAP1 inhibition.**

A  Viability and proliferation of MLS 402-91, MLS 1765-92, and MLS 2645-94 cells cultured in the presence of verteporfin. Data points and error bars represent the mean $\pm$ SEM of one representative experiment performed in quintuplicate.

B  Flow cytometric analysis of apoptosis (cleaved PARP) and mitotic fraction (phosphorylated histone H3$^{S10}$) in MLS cell lines cultured in the presence of 0.25 $\mu$M verteporfin. One of two independent experiments with similar results is shown.

C  Expression of total YAP1 and downstream effectors (FOXM1 and PLK1) in MLS cell lines treated with 0.5 or 1 $\mu$M verteporfin for 15 h. One of at least three independent experiments with similar results is shown.

D  YAP1-responsive luciferase activity in MLS cell lines transfected with a constitutively active YAP1$^{S127A}$ mutant and treated with 1 $\mu$M verteporfin. Relative luciferase activity is displayed relative to control. Bars and error bars represent the mean $\pm$ SD of three independent experiments, unpaired $t$-test.

E  Tumor formation on chicken CAM of MLS cell lines following shRNA-mediated $YAP1$ knockdown. The number of tumors is given above each bar, Fisher exact test; ns, not significant.

F  Tumor growth on chicken CAM of MLS cell lines following treatment with 1 $\mu$M verteporfin. Shown are tumor volumes and representative photographs of tumors. Bars and error bars represent the mean $\pm$ SEM of at least four tumors, unpaired $t$-test.

**Table 1.  IC$_{50}$ values for verteporfin in MLS cell lines.**

| Compound | IC$_{50}$ ($\mu$M) | | |
| --- | --- | --- | --- |
| | MLS 402-91 (type 1) | MLS 1765-92 (type 8) | MLS 2645-94 (type 2) |
| Verteporfin | 0.62 $\pm$ 0.15 | 0.69 $\pm$ 0.10 | 0.82 $\pm$ 0.06 |

Values were calculated by non-linear regression analysis. Results are represented as the mean $\pm$ SEM of at least three independent experiments.

**Table 2.  Fold changes in apoptotic and mitotic fractions of MLS cell lines upon verteporfin treatment.**

| Cell line | Cleaved PARP (D214) | Phosphorylated histone H3 (S10) |
| --- | --- | --- |
| MLS 402-91 (type 1) | 11.58 $\pm$ 0.23 | 0.36 $\pm$ 0.01 |
| MLS 1765-92 (type 8) | 4.00 $\pm$ 0.17 | 0.42 $\pm$ 0.02 |
| MLS 2645-94 (type 2) | 1.72 $\pm$ 0.09 | 0.13 $\pm$ 0.03 |

Results are represented as the mean $\pm$ SEM of at least three independent experiments.

pharmacologic assays need to be interpreted with caution. However, in light of the additional evidence gained from multiple RNAi approaches, we are confident that the effects of verteporfin observed in MLS cells are predominantly due to the inhibition of YAP1-TEAD complexes. Collectively, our data identify YAP1 as a major downstream effector of FUS-DDIT3 in MLS development and document its potential as a novel target for therapeutic intervention.

While YAP1 represents a new player in MLS pathogenesis, alterations of various Hippo signaling intermediates have recently been found in other sarcoma subtypes. A large-scale genomic study described $YAP1$ copy number variations to occur at low frequencies in DDLS and undifferentiated pleomorphic sarcoma/myxofibrosarcoma (Helias-Rodzewicz et al, 2010; Cancer Genome Atlas Research Network, 2017). Furthermore, Eisinger-Mathason and colleagues queried the same dataset to identify copy number losses of $LATS2$ and $SAV1$, effectors of the Hippo signaling cascade that are essential for negative regulation of YAP1 (Eisinger-Mathason et al, 2015). Beyond that, aberrant Hippo signaling can arise from chromosomal translocations involving $YAP1$ or $WWTR1$ (encoding TAZ) as in epithelioid hemangioendothelioma, another fusion gene-driven soft-tissue tumor (Errani et al, 2011; Tanas et al, 2011, 2016; Antonescu et al, 2013). These findings point to a major role of aberrant Hippo signals in different soft-tissue malignancies, which may previously have been underestimated. In MLS, however, genomic alterations affecting the Hippo pathway have not been described (Barretina et al, 2010). Thus, overactive YAP1 signaling in MLS appears to be exclusively mediated by FUS-DDIT3. This insight highlights the potential of functional screening to identify essential genes that evade detection by other genomic technologies and is reminiscent of the situation in alveolar rhabdomyosarcoma, another fusion-driven soft-tissue tumor in which the pathognomonic PAX3-FOXO1 oncoprotein promotes tumorigenesis by dysregulation of YAP1 (Crose et al, 2014). These two entities may therefore define a biologically distinct subgroup of soft-tissue sarcomas in which aberrant YAP1/Hippo signaling is activated by chimeric transcription factors.

We previously described a related and potentially targetable oncogenic mechanism in MLS that involves activation of the IGF1R-PI3K-AKT signaling cascade through FUS-DDIT3-dependent induction of $IGF2$ transcription (Trautmann et al, 2017). Since there is convincing evidence for simultaneous activation of multiple signaling pathways in MLS pathogenesis, future therapeutic concepts need to be focused on an integrated signaling network concept. In this context, comprehensive molecular diagnostic approaches will be key to assign MLS patients to molecularly stratified clinical trials based on the identification of appropriate predictive biomarkers. The data presented here indicate that immunohistochemical screening for nuclear YAP1 could provide such a biomarker, which should be prospectively addressed in future studies.

In conclusion, we have identified dependence on aberrant YAP1 activity as specific liability of FUS-DDIT3-expressing MLS cells, and provide preclinical evidence that YAP1-mediated signal transduction represents a candidate target for therapeutic intervention that warrants further investigation.

# Materials and Methods

### Cell culture

The MLS cell lines MLS 402-91, MLS 1955-91 ($FUS$-$DDIT3$ type 1), MLS 1765-92 (type 8), and MLS 2645-94 (type 2) (all contributed by Pierre Åman) and the liposarcoma cell lines T449, T778 (well-differentiated liposarcoma [WDLS], both kindly provided by Florence Pedeutour), SW872 (obtained from CLS Cell Lines Service), LiSa-2 (pleomorphic liposarcoma [PLS], kindly provided by Silke Brüderlein and Peter Möller), and FU-DDLS-1 (dedifferentiated liposarcoma

[DDLS], kindly provided by Jun Nishio) were cultured in RPMI-1640 supplemented with 10% fetal bovine serum (FBS; Biochrom) and 1% penicillin/streptomycin (P/S; Biochrom). The flasks for culturing MLS 1955-91 were additionally coated with 10% Collagen R solution (Serva). HEK293T cells were kindly provided by William C. Hahn and grown in DMEM (10% FBS, 1% P/S). The SCP-1 cell system (Bocker *et al*, 2008; Haasters *et al*, 2009) was established by Thomas Kindler and cultured in MEM (10% FBS, 1% P/S). Cells were grown under standard conditions (37°C, humidified atmosphere, 5% $CO_2$) and routinely tested for mycoplasma contamination. Cell line identity and purity were verified using the Multiplex Cell Authentification and Contamination Tests (Multiplexion) and/or by documentation of the specific gene fusions. To study the effects of verteporfin (Targetmol dissolved in DMSO [Sigma-Aldrich]), MLS cells were grown in RPMI-1640 with 2% FBS. The final DMSO concentration did not exceed 0.2% for all *in vitro* and *in vivo* applications.

### Vectors and lentiviral transduction

The *YAP1* and *FUS-DDIT3* cDNA were amplified from MLS 402-91 cells. *YAP1* and *FUS-DDIT3* cDNAs were cloned into pLenti6.2/V5-DEST and pLenti7.3/V5-DEST lentiviral expression vectors, respectively, using Gateway Technology (Invitrogen). Short hairpin RNA (shRNA) sequences were taken from the DECIPHER Pooled Lentiviral Human Genome-Wide shRNA Library (Cellecta) and cloned into the BbsI site of the pRSI12-U6-sh-UbiC-TagRFP-2A-Puro or pRSI9-U6-sh-UbiC-TagRFP-2A-Puro lentiviral vectors (Cellecta). shRNA target sequences were as follows: shYAP1-2: 5′-CCC AGT TAA ATG TTC ACC AAT-3′, shYAP1-3: 5′-CAG GTG ATA CTA TCA ACC AA A-3′, shKIF11: 5′-GCG TAC AAG AAC ATC TAT AAT-3′, shEIF3A: 5′-GCG CCT TGA GAG TCT GAA TAT-3′, and shNTC (non-targeting control): 5′-CAA CAA GAT GAA GAG CAC CAA-3′. Generation of viral supernatants and viral transduction was performed as previously described (Scholl *et al*, 2009). Cells were selected with 10 μg/ml blasticidin (Life Technologies) or 2 μg/ml puromycin (Sigma-Aldrich). GFP-positive SCP-1 cells, either transduced with pLenti7.3/V5-DEST-*FUS-DDIT3* or empty vector control, were isolated by means of fluorescent activating cell sorting.

### Short hairpin RNA screening and data analysis

Screens were performed in duplicates using Module 1 of the DECIPHER Pooled Lentiviral Human Genome-Wide shRNA Library (Cellecta), which consists of 27,500 shRNAs targeting 5,043 human genes (preprint: Huellein *et al*, 2018). SCP-1 cells ($3.4 \times 10^7$) stably transduced with either *FUS-DDIT3* or an empty control vector (EV) were transduced with library virus at a multiplicity of infection of 0.7 in the presence of 5 μg/ml polybrene (Millipore). After 3 days, half of the cells were harvested as baseline sample and the other half of the cells were selected with 2 μg/ml puromycin for 3 days, cultured without puromycin for six additional days, and harvested as drop-out sample. Baseline and drop-out samples were subjected to genomic DNA extraction and PCR amplification of barcode regions for high-throughput sequencing as described previously (Słabicki *et al*, 2016).

Raw sequencing data were processed using the DECIPHER Bar-Code Deconvoluter software (Cellecta) for converting read counts of barcode sequences to shRNA read counts. The read counts of individual shRNAs were normalized to the mean of total read counts,

and the $\log_2$ fold change (LFC) was calculated for each cell line by dividing the normalized read counts of the drop-out sample by those of the baseline sample followed by $\log_2$ transformation. LFC values were processed with the GenePattern module "NormLines" using the peak median absolute deviation method (PMAD) (Cheung *et al*, 2011), resulting in rescaled LFC values with similar ranges in all cell lines to obtain comparability. Finally, PMAD normalized values were analyzed by RNAi Gene Enrichment Ranking (RIGER) (Luo *et al*, 2008) to calculate differential gene effects between FUS-DDIT3-expressing SCP-1 cells and other cell lines.

### Short interfering RNA-mediated knockdown

MLS 402-91 and MLS 1765-92 cells were grown in medium supplemented with 2% FBS to a density of 50%, transfected with 25 pmol of pre-validated short interfering RNA (siRNA) targeting *YAP1* (5′-GGA AGG AGA UGG AAU GAA CAU AGA A-3′; Life Technologies, Assay Identifier HSS115944) or a non-targeting control siRNA (BLOCK-iT Alexa Fluor Red Fluorescent Control, Life Technologies) using Lipofectamine RNAiMAX (Life Technologies), and harvested for immunoblotting after 72 h.

### RNA isolation, cDNA synthesis, and quantitative RT–PCR

Total RNA was isolated using the RNeasy Plus Mini Kit (Qiagen) and reverse-transcribed using the High-Capacity cDNA Reverse Transcription Kit (Applied Biosystems). Quantitative RT–PCR was performed on a LightCycler 480 instrument (Roche) using 0.1–100 ng cDNA, 10 μM primers, and LightCycler 480 SYBR Green I Master reagents (Roche) in a total volume of 10 μl. Target gene expression was calculated based on the $\Delta\Delta Ct$ method and normalized to *ACTB* and *GAPDH* as reference genes. Primer sequences are given in Appendix Table S1.

### Tumor specimens and tissue microarrays

Tissue microarrays (TMAs) were prepared from 223 formalin-fixed, paraffin-embedded (with two representative 1-mm cores) liposarcoma specimens (MLS, *n* = 85; WDLS, *n* = 55; DDLS, *n* = 74; PLS, *n* = 9) selected from the archive of the Gerhard-Domagk-Institute of Pathology (Münster, Germany). Diagnoses were reviewed by two experienced pathologists based on current World Health Organization criteria. From each tumor, two areas were selected by two experienced pathologists to account for potential heterogeneity, e.g., with regard to the round cell content of MLS, and occasional necrobiotic areas and their neighborhood were excluded from TMA sampling. The study was approved by the Ethics Review Board of the University of Münster (2015-548-f-S), and experiments were conformed to the principles set out in the World Medical Association Declaration of Helsinki and the United States Department of Health and Human Services Belmont Report. Written informed consent from patients was not requested by the Ethics Review Board of the University of Münster (2015-548-f-S).

### Cell viability and proliferation assays

For RFP competition assays, cells were transduced with lentiviral shRNAs at an efficiency of approximately 50%. The proportion of

RFP-positive cells was measured by flow cytometry on a BD LSR II instrument (BD Biosciences) after 3 days (baseline) and every 2–3 days thereafter. To determine the effects of drug treatment, $2.5 \times 10^3$ MLS cells were seeded in 96-well plates and exposed to increasing concentrations of verteporfin for 72 h. Cell viability and proliferation were measured using the Cell Proliferation Kit I (MTT) (Roche) as previously described (Michels *et al*, 2013; Trautmann *et al*, 2014).

## Immunohistochemistry

Immunohistochemistry (IHC) was performed with a BenchMark ULTRA Autostainer (VENTANA/Roche) on 3-μm TMA sections. The staining procedure included heat-induced (95-100°C) epitope retrieval using Tris-Borate-EDTA buffer (pH 8.4) for 32-72 min, incubation with primary antibodies for 16–120 min, and signal detection using the OptiView DAB IHC Detection Kit (VENTANA/Roche). The following primary antibodies were used: YAP (monoclonal rabbit, D8H1X, 1:100, #14074, Cell Signaling), FOXM1 (monoclonal mouse, G-5, 1:1000, #376471, Santa Cruz), PLK1 (monoclonal rabbit, 208G4, 1:25, #4513, Cell Signaling). Immunoreactivity was assessed using a semi-quantitative score (0, negative; 1, weak; 2, moderate; and 3, strong) defining the staining intensity in the positive control (hepatocellular carcinoma) as strong. Negative control stainings using an appropriate IgG subtype (DCS) were included. Only tumors with at least moderate staining (semi-quantitative score $\geq 2$) and $\geq 30\%$ (YAP1 and FOXM1) or $\geq 5\%$ (PLK1) positive cells were considered positive for the purposes of the study. The IHC readers were blinded to outcome data, and the score cut point (positive = semi-quantitative score $\geq 2$) was pre-specified without prior analyses of the clinical course.

## Immunofluorescence

Cells were grown on poly-L-lysine-coated chamber slides (Sigma-Aldrich) or collagen-coated coverslips (Corning), fixed in 4% paraformaldehyde (PFA), permeabilized and blocked with 5% BSA or 2% goat serum and 0.3-3% Triton X-100, and incubated with primary antibodies (YAP, monoclonal rabbit, D8H1X, 1:100, #14074, Cell Signaling; CHOP/DDIT3, monoclonal mouse, #2895, Cell Signaling, 1:1,000) at 4°C over night. After secondary antibody incubation (Life Technologies, 1:1,000), cells were mounted in Vectashield Mounting Medium with DAPI (Vector Laboratories) or ProLong Gold Antifade Reagent with DAPI (Cell Signaling). Immunofluorescence (IF) analysis was performed with a Leica DM5500 B microscope or a Leica TCS SP5 confocal microscope. Co-localization analysis was performed using Fiji software plugins "Colocalization Finder" and "Image Correlator" to obtain correlation plots and calculate Pearson correlation coefficients (Schindelin *et al*, 2012).

## Immunoblotting

Cytoplasmic and nuclear fractions were prepared from cell pellets with 0.1% NP-40 lysis buffer (0.1% NP-40 [BioVision] and Halt Protease and Phosphatase Inhibitor Cocktail [Thermo Fischer, 1:100] in Dulbecco's phosphate-buffered saline [DPBS]) to obtain the cytoplasmic fraction, followed by nuclear lysis with RIPA buffer (50 mM Tris–HCl, 150 mM NaCl, 0.1% SDS, 0.5% sodium deoxycholate, 1%

Triton X-100, Halt Protease and Phosphatase Inhibitor Cocktail [1:100]) to obtain the nuclear fraction. Whole-cell lysates were prepared with Triton X-100 lysis buffer (20 mM Tris–HCl, 150 mM NaCl, 1 mM EDTA, 10% Glycerol, 1% Triton X-100, Halt Protease and Phosphatase Inhibitor Cocktail [1:100]). Protein extracts (10–50 μg) were subjected to SDS–PAGE and transferred to PVDF membranes (Carl Roth). Membranes were blocked with 5% dry milk in TBST, followed by incubation with primary and HRP-linked secondary antibodies. Chemiluminescent signals were detected by autoradiography or the Molecular Imager ChemiDoc System (Bio-Rad). Signal intensities were quantified using Fiji software (Schindelin *et al*, 2012). Antibodies are given in Appendix Table S2.

## Immunoprecipitation

HEK293T cells were transfected with *YAP1* and *FUS-DDIT3* expression plasmids (pLenti6.2/V5-DEST and pLenti7.3/V5-DEST, respectively) and lysed with Triton X-100 lysis buffer after 40 h. Protein lysates (500 μg) were incubated with 30 μl SureBeads Protein G Magnetic Beads (Bio-Rad) and 1 μl V5 Tag Monoclonal Antibody (Invitrogen) in a total volume of 200 μl. MLS 1765-92 cells were washed twice with 0.1% NP-40 lysis buffer to remove the cytoplasm, and nuclei were resuspended in HEPES lysis buffer (25 mM HEPES, 100 mM NaCl, 1 mM EDTA, 0.5% Triton X-100, Halt Protease and Phosphatase Inhibitor Cocktail [1:100]) and sonicated using a Diagenode Bioruptor device. After centrifugation, protein supernatants (750 μg) were incubated with 30 μl SureBeads Protein G Magnetic Beads and 4 μl YAP antibody (#14074, Cell Signaling, 1:50) or 4 μl DDIT3 antibody (#2895, Cell Signaling, 1:50) in a total volume of 200 μl, and 1 μg rabbit IgG (sc-2027, Santa Cruz) and 1 μg mouse IgG2a (554126, BD Biosciences) served as controls. Beads were washed and subjected to immunoblotting on the next day.

## Luciferase assays

To assess YAP1-mediated transcriptional activity, MLS cells were transfected with TEAD (8xGTIIC) luciferase reporter plasmid (Dupont *et al*, 2011). For extrinsic activation of YAP1, MLS cells were co-transfected with a constitutively active YAP1$^{S127A}$ mutant (Zhao *et al*, 2007). The amount of plasmid DNA in each transfection was kept constant by addition of the non-coding plasmid backbone. Reporter assays were performed in triplicates using the Dual-Luciferase Reporter Assay System (Promega) after 24 h. Firefly luciferase activity was normalized to a co-transfected Renilla pRL-TK control plasmid (Promega) to account for differences in transfection efficiency. For verteporfin treatment, medium containing transfection reagent was replaced after 6 h with medium containing 1 μM verteporfin and 2% FBS.

## Cell cycle and apoptosis analysis

Cell cycle analysis was performed with the Click-iT Plus EdU Flow Cytometry Assay Kit (Invitrogen) and DAPI as DNA staining reagent (BD Biosciences). Five days after shRNA transduction, cells were incubated with 10 μM EdU solution for 1 h, washed with 3 ml 1% BSA/DPBS, fixed with 4% PFA, permeabilized, and incubated with EdU detection solution and DAPI. Stained cells were acquired on a BD LSR II flow cytometer (BD Biosciences) within 1 h.

The effects of verteporfin on apoptotic and mitotic rates were assessed by flow cytometric detection of cleaved PARP and phosphorylated histone H3$^{S10}$, respectively. MLS cells grown in medium supplemented with 2% FBS were treated with 0.25 μM verteporfin for 72 h, detached using 0.025% trypsin (Life Technologies), fixed in 2% PFA, washed in PBS, and permeabilized in 0.25% Triton X-100/PBS for 5 min on ice. After an additional washing step, cells were stained for 60 min with PE Mouse anti-Cleaved PARP (Asp214) (BD Biosciences) and phospho-Histone H3$^{S10}$ (#9716, Alexa Fluor 647 Conjugate, Cell Signaling) antibodies. Fluorescence intensity was measured using a FACSCanto II flow cytometer, and data were analyzed using the FACSDiva software (BD Biosciences).

### Senescence assays

Five days after shRNA transduction, cells were fixed with 4% PFA and stained with the Senescence β-Galactosidase Staining Kit (Cell Signaling). Images of 10 random fields per sample were taken with a Zeiss Cell Observer microscope at ×100 magnification. The staining intensity of senescent cells was determined using Fiji software (Schindelin *et al*, 2012). Normalized staining intensities were calculated dividing the total intensity by the number of β-galactosidase-positive cells.

### Chicken chorioallantoic membrane assays

MLS 402-91 and MLS 1765-92 cells were transduced with lentiviral shRNAs, selected with 2 μg/ml puromycin for 2 days, and after four additional days, $1.5 \times 10^6$ cells were deposited within 5-mm silicon rings on the surface of chicken chorioallantoic membranes (CAM) 8 days postfertilization. Images were acquired after 4 days of incubation, and tumor areas were calculated using Fiji software and normalized to the area of the silicon ring.

For drug treatment, CAM assays were performed as previously described (Syrovets *et al*, 2005). In brief, $1 \times 10^6$ MLS 402-91 and MLS 1765-92 cells in medium and Matrigel (1:1) were xenografted onto chicken CAM (within 5-mm silicon rings), and 1 μM verteporfin or vehicle (0.2% DMSO in NaCl 0.9%) was applied topically on days 8 and 9. Three days after treatment initiation, xenografts were explanted, fixed in 5% PFA, and processed for histopathological examination. Tumor volume (mm$^3$) was calculated according to the formula: length (mm) × width$^2$ (mm) × π/6 (Tomayko & Reynolds, 1989).

### Statistical analysis

Statistical analysis was performed using paired or unpaired two-tailed *t*-test, two-way ANOVA, or Fisher exact test as appropriate. *P*-values < 0.05 were considered significant. Computations were performed using GraphPad Prism (GraphPad Software).

**Expanded View** for this article is available online.

## Acknowledgements

The authors thank Charlotte Sohlbach, Inka Buchroth, Christian Bertling, Damir Krunic, and the DKFZ Imaging and Cytometry Core Facility for excellent technical support. Florence Pedeutour (Laboratory of Solid Tumor Genetics, Institute for Research on Cancer and Aging of Nice; Laboratory of Solid

### The paper explained

#### Problem

Myxoid liposarcoma (MLS) is an aggressive mesenchymal malignancy with few therapeutic options. Most MLS are driven by the *FUS-DDIT3* fusion gene. However, the mechanisms underlying MLS development, including clinically actionable genetic vulnerabilities, are incompletely understood.

#### Results

Pooled RNA interference (RNAi) screening uncovered context-dependent essentiality of *YAP1*, encoding a transcriptional co-activator, in FUS-DDIT3-expressing mesenchymal stem cells. Immunohistochemistry analysis of MLS patient specimens revealed that nuclear YAP1 expression is significantly more prevalent in MLS compared to other liposarcoma subtypes. *YAP1* depletion in MLS cell lines caused suppression of cell viability, cell cycle arrest, cellular senescence, and induction of apoptosis accompanied by decreased YAP1 target gene expression, and primary MLS tumors showed strong expression of YAP1 downstream effectors. Mechanistically, FUS-DDIT3 promoted *YAP1* transcription, nuclear localization, and transcriptional activity and physically associated with YAP1 in the nucleus of MLS cells. Pharmacologic inhibition of YAP1 activity with verteporfin suppressed cell viability and YAP1 target gene expression in MLS cell lines, and the growth-inhibitory effects of *YAP1* knockdown or verteporfin treatment could be recapitulated in MLS cell line-based xenograft models.

#### Impact

These findings provide insight into the functional underpinnings of MLS development. More broadly, the data underscore the potential of functional screens for uncovering vulnerabilities in cancers with low mutational burden, such as translocation-related sarcomas, whose critical dependencies evade detection by DNA and RNA sequencing. In addition, these results may have implications for the nascent field of "precision sarcoma medicine". On the one hand, the "druggability" of YAP1 suggests a rational strategy to selectively target FUS-DDIT3-expressing MLS cells. Secondly, nuclear YAP1 expression may represent a biomarker to identify MLS patients that could benefit from a YAP1-directed therapeutic approach within future clinical trials.

Tumor Genetics, University Hospital of Nice-Côte d'Azur University, Nice, France) kindly provided the T449 and T778 cell lines. Silke Brüderlein and Peter Möller (Institute of Pathology, Ulm University, Ulm, Germany) kindly provided the LiSa-2 cell line. Jun Nishio (Faculty of Medicine, Department of Orthopaedic Surgery, Fukuoka University, Fukuoka, Japan) kindly provided the FU-DDLS-1 cell line. William C. Hahn (Dana-Farber Cancer Institute, Boston, USA) kindly provided the HEK293T cell line. This study was supported in part by grants from the Deutsche Forschungsgemeinschaft (W. Hartmann and M. Trautmann, HA4441/2-1; N. Azoitei, AZ96/1-3), the Wilhelm Sander-Stiftung (W. Hartmann and M. Trautmann, 2016.099.1), and the "Innovative Medizinische Forschung" funding program of Münster University Medical School (M. Trautmann, TR121716 and I-TR221611; M. Trautmann and S. Huss, I-HU121421). Y.-Y. Cheng was supported by a stipend from the Helmholtz International Graduate School for Cancer Research. A. Ståhlberg was supported by the Swedish Cancer Society (grant 2016-438) and the Swedish Research Council (grant 2017-01392).

## Author contributions

Conception and design of the study: MT, Y-YC, CS, WH, SF. Acquisition, analysis, and/or interpretation of data: MT, Y-YC, PJ, NA, IB, JH, MS, II, MC, RB, BA, SH, TZ, UL, TK, CS, WH, SF. Contribution of administrative, experimental, analytic

or material support: EW, SH, CR, TS, AS, PÅ. Writing of the manuscript: MT, Y-YC, CS, WH, SF. Study supervision: MT, CS, WH, SF.

## Conflict of interest

The authors declare that they have no conflict of interest.

## For more information

(i) Division of Translational Pathology, GDI—Cancer Research Lab
https://www.ukm.de/index.php?id=gdicancerresearchlab

(ii) Department of Translational Medical Oncology, NCT Heidelberg and DKFZ
https://www.nct-heidelberg.de/en/tmo

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
