## [Review Process File · EMBO Molecular Medicine]

Requirement for YAP1 signaling in myxoid liposarcoma

Marcel Trautmann, Ya-Yun Cheng, Patrizia Jensen, Ninel Azoitei, Ines Brunner, Jennifer Hüllein, Mikolaj Slabicki, Ilka Isfort, Magdalene Cyra, Ruth Berthold, Eva Wardelmann, Sebastian Huss, Bianca Altwater, Claudia Rossig, Susanne Hafner, Thomas Simmet, Anders Ståhlberg, Pierre Åman, Thorsten Zenz, Undine Lange, Thomas Kindler, Claudia Scholl, Wolfgang Hartmann & Stefan Fröhling

Review timeline:

Submission date:	31 October 2018
Editorial Decision:	13 December 2018
Revision received:	30 January 2019
Editorial Decision:	12 February 2019
Revision received:	21 February 2019
Accepted:	25 February 2019

Editor: Lise Roth

Transaction Report:

1st Editorial Decision

13 December 2018

Thank you for the submission of your manuscript to EMBO Molecular Medicine, and my apologies for the delay in getting back to you, which is due to the late review from one referee. We have now heard back from the three referees who were asked to evaluate your manuscript. As you will see from the reports below, they are overall positive and support publication of the article in EMBO Molecular Medicine pending appropriate revisions. They however highlight the need to further strengthen the study through minor additional experiments, and more thorough discussion and clarifications. Mouse experiments (mentioned by reviewer #3) will not be required for acceptance of the manuscript. EMBO Molecular Medicine encourages a single round of revision only and therefore, acceptance or rejection of the manuscript will depend on the completeness of your responses included in the next, final version of the manuscript.

EMBO Molecular Medicine has a "scooping protection" policy, whereby similar findings that are published by others during review or revision are not a criterion for rejection. Should you decide to submit a revised version, I do ask that you get in touch after three months if you have not completed it, to update us on the status. Please also contact us as soon as possible if similar work is published elsewhere. If other work is published, we may not be able to extend the revision period beyond three months.

I look forward to receiving your revised manuscript.

***** Reviewer's comments *****

Referee #1 (Comments on Novelty/Model System for Author):

This is an excellent paper based on robust methodology. Research has been undertaken using variety of approaches that seems to be coherent. Non ethical issues should be raised.

Referee #1 (Remarks for Author):

This is an excellent paper demonstrating the role of YAP in myxoid liposarcoma molecular oncogenesis. As authors suggest that YAP involvement may be limited to MLPS they may briefly comment how should be interpreted YAP up-regulation in approximately one third of dedifferentiated liposarcoma.

Referee #2 (Remarks for Author):

Trautman et al., explore genes required for survival of myxoid liposarcoma cells, which are transformed by the FUS-DDIT3 transcription factor fusion. Using an shRNA screen, they identify YAP as the top target and then perform a range of in vitro, patient sample and transcriptional studies that go some way to validating YAP as an important downstream gene relevant for myxoid liposarcoma. They also provide hints at mechanism by showing that YAP and FUS-DDIT3 can form a physical complex. In order to fully validate that these proteins operate together to regulate transcription it would require substantially more work, but I do not deem that such work is necessary for the current manuscript.

In summary, the study is generally well performed and has interesting findings that will spur future studies. I have some minor comments aimed at improving the manuscript.

Minor comments - it would be valuable for readers to know more about the results of the screen, e.g. by providing a full gene list. This would enable readers to, for example, see where other cancer genes and Hippo pathway genes other than YAP scored in the screen.

The legend for Figure 2 is inaccurate: "Increased YAP1 activity in FUS-DDIT3-expressing mesenchymal stem cells, MLS cell lines, and MLS patient samples." YAP expression and subcellular localization, but not activity, is assessed. Nuclear YAP/Yorkie does not always correspond with its activity.

The co-IPs in Figure 4E are missing an important control. The V5 pulldown should be performed in cells expressing FUS-DDIT3 but not V5-YAP and then the V5 pulldown performed to determine whether FUS-DDIT3 can be non-specifically precipitated by anti-V5.

Care should be taken when interpreting the verteporphin studies (cell line and chicken CAM) and this should be reflected in the manuscript as the drug has cytotoxic effects and in some settings these have been shown to be independent of YAP, TAZ and TEADs.

Referee #3 (Comments on Novelty/Model System for Author):

The paper is extremely well written. The experiments are well controlled and well presented. Findings from this paper establish a new target for therapeutic intervention in a rare, aggressive form of cancer.

Referee #3 (Remarks for Author):

Trautmann et al identify YAP1, a central component of the Hippo pathway, as an important oncogenic effector in mesenchymal cells expressing the FUS:DDIT3 fusion oncogene. They show increased YAP1 activity in myxoid liposarcoma tissue and propose that YAP1 and FUS:DDIT3 interact to coordinate a gene expression program that promotes myxoid liposarcoma malignancy. Finally, they demonstrate that Verteporfin, which inhibits YAP1, inhibits myxoid liposarcoma growth in cell lines and chicken CAM.

The paper is extremely well written. The experiments are well controlled and well presented. Findings from this paper establish a new target for therapeutic intervention in a rare, aggressive form of cancer.

The paper does not address how YAP1 and FUS:DDIT3 interact to promote myxoid liposarcoma malignancy. Also, it would be nice to see mouse data on the potential efficacy of Verteporfin against myxoid liposarcoma. The Verteporfin doses used to achieve anti-tumor effects are fairly high; it may be difficult to achieve effective levels in vivo.

I expect that the authors will direct their attention towards mechanism of actions and further preclinical validation in future studies.

I recommend acceptance of the paper.

1st Revision - authors' response

30 January 2019

We thank the Reviewers and the Editor for their insightful and constructive comments, which have substantially improved this work. Please find enclosed a revised manuscript that has been modified in accordance with their recommendations. Our specific responses to the Reviewers' comments are detailed individually below.

Referee #1 (Comments on Novelty/Model System for Author):

This is an excellent paper based on robust methodology. Research has been undertaken using variety of approaches that seems to be coherent. Non ethical issues should be raised.

We are grateful for these favorable comments.

Referee #1 (Remarks for Author):

This is an excellent paper demonstrating the role of YAP in myxoid liposarcoma molecular oncogenesis. As authors suggest that YAP involvement may be limited to MLPS they may briefly comment how should be interpreted YAP up-regulation in approximately one third of dedifferentiated liposarcoma.

We thank the Reviewer for this helpful question. Indeed, the wording of the abstract implied that YAP1 upregulation represents a unique feature of myxoid liposarcoma (MLS), and the Reviewer is right in pointing out that a subset of dedifferentiated liposarcoma (DDL) also express YAP1 (Figure 2E and F), albeit to a much lesser extent. This finding is in agreement with data by Fullenkamp et al. (Oncotarget 7:30094-108, 2016). While our findings provide functional and mechanistic evidence that FUS-DDIT3 drives YAP1 expression in MLS, Eisinger-Mathason et al. queried the TCGA dataset to identify DNA copy number loss of *LATS2* and *SAVI*, encoding essential negative regulators of YAP1, in a relevant proportion of high-grade sarcomas, including DDL (Proc Natl Acad Sci USA 112:E3402-11, 2015). Thus, genetic deletion of effectors of the Hippo signaling cascade may represent an alternative mechanism of pathway deregulation that underlies elevated YAP1 expression in other liposarcoma subtypes. Furthermore, we previously identified aberrant activation of the cytoplasmic tyrosine kinase SRC in a subset of well-differentiated liposarcoma/DDL (Sievers et al. Int J Cancer 137:2578-88, 2015), and Reuven and colleagues recently documented that SRC may contribute to YAP1/TAZ activation (J Biol Chem pii: jbc.RA118.004364, 2018). To reflect this complex situation, we have modified the somewhat misleading wording of the abstract (page 2; line 57).

Referee #2 (Remarks for Author):

Trautman et al., explore genes required for survival of myxoid liposarcoma cells, which are transformed by the FUS-DDIT3 transcription factor fusion. Using a shRNA screen, they identify YAP as the top target and then perform a range of in vitro, patient sample and transcriptional studies that go some way to validating YAP as an important downstream gene relevant for myxoid liposarcoma. They also provide hints at mechanism by showing that YAP and FUS-DDIT3 can form a physical complex. In order to fully validate that these proteins operate together to regulate transcription it would require substantially more work, but I do not deem that such work is necessary for the current manuscript. In summary, the study is generally well performed and has interesting findings that will spur future studies. I have some minor comments aimed at improving the manuscript.

We are delighted that the Reviewer found this a well-performed study that has yielded interesting results and will stimulate future investigations.

Minor comments - it would be valuable for readers to know more about the results of the screen, e.g. by providing a full gene list. This would enable readers to, for example, see where other cancer genes and Hippo pathway genes other than YAP scored in the screen.

We agree and have included the results of the RNA interference (RNAi) screen conducted in human mesenchymal stem cells in the revised manuscript. Specifically, we now provide the full gene list generated by RNAi Gene Enrichment Ranking, which was employed to rank genes with respect to FUS-DDIT3-selective essentiality, as Appendix Table S1 (page 4; lines 115-116).

The legend for Figure 2 is inaccurate: "Increased YAP1 activity in FUS-DDIT3-expressing mesenchymal stem cells, MLS cell lines, and MLS patient samples." YAP expression and subcellular localization, but not activity, is assessed. Nuclear YAP/Yorkie does not always correspond with its activity.

We agree and have amended the legend to Figure 2 (page 24; line 772).

The co-IPs in Figure 4E are missing an important control. The V5 pulldown should be performed in cells expressing FUS-DDIT3 but not V5-YAP and then the V5 pulldown performed to determine whether FUS-DDIT3 can be non-specifically precipitated by anti-V5.

Thank you for this helpful suggestion. We have performed this experiment and found that FUS-DDIT3 was not precipitated by the anti-V5 antibody (Figure 4E).

Care should be taken when interpreting the verteporfin studies (cell line and chicken CAM) and this should be reflected in the manuscript as the drug has cytotoxic effects and in some settings these have been shown to be independent of YAP, TAZ and TEADs.

We completely agree with the Reviewer that the specificity of verteporfin is limited. Verteporfin was initially described as a photosensitizer that is used in photodynamic therapy of neovascular macular degeneration, where it is activated by laser light to generate reactive oxygen species targeting abnormal blood vessels (Miller et al. Arch Ophthalmol 117:1161-73, 1999). Based on a large-scale drug screen, Liu-Chittenden and colleagues were the first to identify verteporfin's function as an inhibitor of TEAD-YAP1 association (Genes Dev 26:1300-5, 2012). Since then, verteporfin has been employed in multiple studies as a prototype small-molecule inhibitor of the YAP1-TEAD interaction (e.g. Weiler et al. Gastroenterology 152:2037-51, 2017; Tranchant et al. Clin Cancer Res 23:3191-202, 2017; Yu et al. Cancer Cell 25:822-30, 2014). Zhang et al. described inhibitory effects of verteporfin in different colon cancer models that were not related to blockade of YAP1 or TAZ activity but due to impaired clearance of high-molecular-weight protein aggregates (Sci Signal 8:ra98, 2015). Thus, it is likely that verteporfin can affect cancer cells through several mechanisms. However, given the concordance between our pharmacologic data and the results obtained by various genetic approaches, we are confident that the effects of verteporfin in MLS cells are primarily due to YAP1-TEAD complex inhibition. Nonetheless, we agree with the Reviewer that a more nuanced discussion of the drug's properties is warranted, and we have amended the Discussion section accordingly (page 9; lines 276-281).

Referee #3 (Comments on Novelty/Model System for Author):

The paper is extremely well written. The experiments are well controlled and well presented. Findings from this paper establish a new target for therapeutic intervention in a rare, aggressive form of cancer.

Thank you for these positive and encouraging comments. We share the Referee's view that YAP1 represents a novel therapeutic target that warrants further study to improve outcomes for patients with MLS.

Referee #3 (Remarks for Author):

Trautmann et al identify YAP1, a central component of the Hippo pathway, as an important oncogenic effector in mesenchymal cells expressing the FUS:DDIT3 fusion oncogene. They show increased YAP1 activity in myxoid liposarcoma tissue and propose that YAP1 and FUS:DDIT3 interact to coordinate a gene expression program that promotes myxoid liposarcoma malignancy. Finally, they demonstrate that Verteporfin, which inhibits YAP1, inhibits myxoid liposarcoma growth in cell lines and chicken CAM.

The paper is extremely well written. The experiments are well controlled and well presented. Findings from this paper establish a new target for therapeutic intervention in a rare, aggressive form of cancer.

The paper does not address how YAP1 and FUS:DDIT3 interact to promote myxoid liposarcoma malignancy. Also, it would be nice to see mouse data on the potential efficacy of Verteporfin against myxoid liposarcoma. The Verteporfin doses used to achieve anti-tumor effects are fairly high; it may be difficult to achieve effective levels *in vivo*. I expect that the authors will direct their attention towards mechanism of actions and further preclinical validation in future studies. I recommend acceptance of the paper.

We are grateful for these favorable comments. As pointed out by the Reviewer, the mechanistic details of the interaction between FUS-DDIT3 and YAP1 need to be deciphered in future studies. Our current hypothesis is that FUS-DDIT3 and YAP1 interact in chromatin-associated multi-protein complexes whose analysis will require extensive proteomic experimentation.

We are aware of the limitations of the avian chorioallantoic membrane (CAM) model as an *in vivo* tool. However, the only patient-derived xenograft model reported thus far (Qi et al. Oncotarget 8:54320-30, 2017) is not available to us, and previous attempts to establish and propagate MLS cell line-based xenotransplants have, to our best knowledge, not been successful. We will continue our efforts to establish a mammalian *in vivo* system for future experiments.

2nd Editorial Decision

12 February 2019

Thank you for the submission of your revised manuscript to EMBO Molecular Medicine. We have now received the enclosed reports from the referees. As you will see the reviewers are now supportive, and I am pleased to inform you that we will be able to accept your manuscript pending minor editorial amendments.

***** Reviewer's comments *****

Referee #1 (Remarks for Author):

Authors have adequately dealt with reviewer's remarks.

Referee #2 (Remarks for Author):

The authors have responded to all of the reviewer queries, which were on the minor end of the scale.

Referee #3 (Remarks for Author):

The reviewers' concerns have been addressed, and I recommend acceptance of the paper.

2nd Revision - authors' response

21 February 2019

Authors made the requested changes.

Corresponding Author Name: Wolfgang Hartmann and Stefan Fröhling

Manuscript Number: EMM-2018-09889-V2